# HIF-Overexpression and Pro-Inflammatory Priming in Human Mesenchymal Stromal Cells Improves the Healing Properties of Extracellular Vesicles in Experimental Crohn’s Disease

**DOI:** 10.3390/ijms222011269

**Published:** 2021-10-19

**Authors:** Marta Gómez-Ferrer, Elena Amaro-Prellezo, Akaitz Dorronsoro, Rafael Sánchez-Sánchez, Ángeles Vicente, Jesús Cosín-Roger, María Dolores Barrachina, María Carmen Baquero, Jaris Valencia, Pilar Sepúlveda

**Affiliations:** 1Regenerative Medicine and Heart Transplantation Unit, Instituto de Investigación Sanitaria La Fe, 46026 Valencia, Spain; margofe22@gmail.com (M.G.-F.); elenaamaro23@gmail.com (E.A.-P.); akaitz82@gmail.com (A.D.); rafa_4586@hotmail.com (R.S.-S.); 2Sección Departamental de Biología Celular, Facultad de Medicina, Universidad Complutense de Madrid, 28040 Madrid, Spain; avicente@ucm.es; 3Instituto de Investigación Sanitaria, Hospital Clínico San Carlos, 28040 Madrid, Spain; 4Hospital Dr Peset, FISABIO, 46017 Valencia, Spain; jesus.cosin@uv.es; 5Departamento de Farmacología, Facultad de Medicina, University of Valencia, 46010 Valencia, Spain; dolores.barrachina@uv.es; 6Servicio de Cirugía Oral y Maxilofacial, Hospital Universitari I Politècnic La Fe, Avenida Fernando Abril Martorell, 106, 46026 Valencia, Spain; baquero_car@gva.es

**Keywords:** mesenchymal stromal cells, extracellular vesicles, hypoxia-inducible factor 1-alpha, immunomodulation, macrophage repolarization, Crohn’s disease

## Abstract

Extracellular vesicles (EVs) derived from mesenchymal stromal cells (MSCs) have therapeutic potential in the treatment of several immune disorders, including ulcerative colitis, owing to their regenerative and immunosuppressive properties. We recently showed that MSCs engineered to overexpress hypoxia-inducible factor 1-alpha and telomerase (MSC-T-HIF) and conditioned with pro-inflammatory stimuli release EVs (EV_MSC-T-HIF_^C^) with potent immunomodulatory activity. We tested the efficacy of EV_MSC-T-HIF_^C^ to repolarize M1 macrophages (Mφ1) to M2-like macrophages (Mφ2-like) by analyzing surface markers and cytokines and performing functional assays in co-culture, including efferocytosis and T-cell proliferation. We also studied the capacity of EV_MSC-T-HIF_^C^ to dampen the inflammatory response of activated endothelium and modulate fibrosis. Finally, we tested the therapeutic capacity of EV_MSC-T-HIF_^C^ in an acute colitis model. EV_MSC-T-HIF_^c^ induced the repolarization of monocytes from Mφ1 to an Mφ2-like phenotype, which was accompanied by reduced inflammatory cytokine release. EV_MSC-T-HIF_^c^-treated Mφ1 had similar effects of immunosuppression on activated peripheral blood mononuclear cells (PBMC) as Mφ2, and reduced the adhesion of PBMCs to activated endothelium. EV_MSC-T-HIF_^c^ also prevented myofibroblast differentiation of TGF-β-treated fibroblasts. Finally, administration of EV_MSC-T-HIF_^c^ promoted healing in a TNBS-induced mouse colitis model in terms of preserving colon length and intestinal mucosa architecture and altering the ratio of Mφ1/ Mφ2 infiltration. In conclusion, EV_MSC-T-HIF_^C^ have effective anti-inflammatory properties, making them potential therapeutic agents in cell free-based therapies for the treatment of Crohn’s disease and likely other immune-mediated inflammatory diseases.

## 1. Introduction

Mesenchymal stromal cells (MSCs) exert immunomodulatory and tissue repair effects that promote regenerative processes, making them potentially useful in the treatment of several different clinical pathologies [1,2,3]. Owing to their robust immunosuppressive potential, MSCs have been used to treat several immune-related diseases such as graft-versus-host disease [3,4], systemic lupus erythematosus and Crohn’s disease [5]. There is a wealth of evidence indicating that the therapeutic benefits of MSCs are mainly attributed to paracrine activity, including intercellular signaling events mediated by membrane-enclosed extracellular vesicles (EVs) [6]. The responsiveness of MSCs is known to be influenced by microenvironmental cues in injured tissues [7]. In this regard, the production of MSCs using standard culture conditions can lead to insufficient therapeutic responses when used in patients, likely due to poor engraftment, low survival and limited paracrine effect [8]. One strategy to boost the therapeutic potential of MSCs is to precondition the cells ex vivo using culture conditions that mimic the injured environment [9]. Early stages of the tissue healing process are characterized by hypoxia and a strong inflammatory response [10,11], and pre-exposure of MSCs to hypoxia and/or inflammatory cytokines has proven to be a good strategy to enhance their regenerative potential [12]. In this line, we previously reported that overexpression of hypoxia inducible factor-1α (HIF-1α), a key gene in the adaptive response to hypoxia, in dental pulp-derived MSCs potentiates the therapeutic capacity of both the parental cells and their secreted EVs [13,14]. Other authors have shown that HIF-1α overexpression in bone marrow MSC-derived exosomes induces angiogenesis and reduces fibrosis in a rat model of myocardial infarction [15]. We recently reported that HIF-1α overexpressing MSCs preconditioned with an inflammatory cytokine cocktail secreted EVs (EV_MSC-T-HIF_^C^) with augmented immunoregulatory capacity [16]. The application of EV_MSC-T-HIF_^C^ suppressed activated T-cell proliferation effectively in vitro and significantly decreased tissue swelling in a delayed-type hypersensitivity mouse model by inhibiting cellular infiltration and improving tissue integrity [16]. 

MSC-derived EVs (MSC-EVs) can recapitulate the immunoregulatory effect of MSCs by targeting different immune cell populations, including macrophages (Mφ) [17]. Mφ play essential roles in innate and adaptive immunity and homeostasis, and their inherent functional plasticity allows them to respond to different stimuli. Traditionally (although likely oversimplified), Mφ exhibit two opposite but complementary activation states: classical Mφ1 and alternatively activated Mφ2. The former are considered as pro-inflammatory Mφ and are characterized by the expression of pro-inflammatory cytokines, whereas the latter dampen inflammation, promote the regeneration and healing of tissues and have immunomodulatory functions [18]. 

Beyond immune cells, endothelial cells and fibroblasts are also active key elements in the inflammatory process, as vascular endothelium controls the adhesion and migration of inflammatory cells and is sensitized by inflammatory factors [19]. Specifically, fibroblasts can modulate immune cell behavior by conditioning the environment and regulating the deposition of extracellular matrix (ECM) in injured tissues [20]. Moreover, fibroblasts can remain activated after the acute inflammatory phase is complete, contributing to the development of chronic inflammation and fibrosis [21].

In the present study, we sought to investigate the effects of MSC-EVs on cells involved in the immune response and inflammation. Specifically, we tested the immunoregulatory potential of “boosted” EV_MSC-T-HIF_^C^ for Mφ1 repolarization to Mφ2, and also their effect on endothelium and fibroblast activation in an inflammatory context. We show that administration of EV_MSC-T-HIF_^C^ to mice with experimental acute colitis induced by 2,4,6-trinitrobenzene sulfonic acid (TNBS) accelerates mucosal healing and reduces inflammation and intestinal fibrosis. 

## 2. Results

### 2.1. Characterization of EVs

We isolated EVs from the conditioned medium of human dental pulp MSCs by sequential centrifugation and filtration. MSCs were first immortalized with human telomerase enzyme (named MSC-T) to standardize the culture and purification conditions. Additionally, we overexpressed HIF-1α in MSCs to bolster their regenerative and immunoregulatory capacity (named MSC-T-HIF). MSC-T and MSC-T-HIF have been characterized in of MSC-T and MSC-T-HIF was tested in a previous study [16]. Prior to EV isolation, MSCs were also preconditioned using a cocktail of cytokines that favor their functional activity in an inflammatory context (EV_MSC-T_^C^ and EV_MSC-T-HIF_^C^; the C-superscript denotes cytokine preconditioning) [17]. The diameter of the purified EVs was 150–200 nm, as determined by nanoparticle tracking analysis (NTA) (Figure 1A). Western blot analysis revealed that the EVs expressed the typical exosome markers Hsp70, TSG101 and CD9 and were negative for the endoplasmic reticulum protein calnexin (Figure 1B). Transmission electron microscopy (TEM) analysis of EVs revealed a cup-shaped morphology that is typically observed with the ultracentrifugation protocol, and the size was consistent with the observed NTA-measured diameter (Figure 1C). Finally, EVs showed positive immunogold-labeling with the canonical EV marker CD63 (Figure 1C).

### 2.2. EV_MSC-T_^C^ and EV_MSC-T-HIF_^C^ Are Internalized by Different Leukocyte Subpopulations

We investigated the capacity of different immune cells to internalize EVs. Equal amounts of EVs from two MSC lines (EV_MSC-T_^C^ and EV_MSC-T-HIF_^C^) were labeled with carboxyfluorescein succinimidyl ester (CFSE) and EV uptake was assessed after their addition to different leukocyte subpopulations. Internalization was quantified by flow cytometry using specific markers for T-cells (CD4^+^/CD8^+^), monocytes (CD14^+^), B-cells (CD19^+^), natural killer (NK) cells (CD56^+^) and neutrophils (CD15^+^). Results showed that all immune cell populations tested internalized MSC-EVs, albeit with different efficiency (Figure 1D), suggesting an active EV-based communication mechanism between MSCs and immune cells (Figure 1D and Appendix A). Monocytes and neutrophils showed the strongest EV uptake capacity (50–60%) and no differences were observed in uptake between EV_MSC-T_^C^ and EV_MSC-T-HIF_^C^.

### 2.3. EV_MSC-T-HIF_^C^ Repolarize Mφ1 to an Mφ2-like Phenotype More Efficiently than EV_MSC-T_^C^

An immunomodulatory role of MSCs in the polarization of Mφ2 has been reported [22]. We developed an assay to study the effect of EVs on monocyte cultures differentiated to Mφ1 or Mφ2 using GM- or M-CSF, respectively (Figure 2A). At day 5 of differentiation, and prior to activation, we observed that Mφ1 that were differentiated in the presence of EVs morphologically resembled Mφ2 (Figure 2B). We used flow cytometry to assess the percentage of cells that were CD14^+^CD163^+^, a classical Mφ2 phenotype, finding that the addition of EVs significantly increased the percentage of CD14^+^CD163^+^ cells with respect to Mφ1 differentiated in their absence (Figure 2C). The repolarization capacity of EV_MSC-T-HIF_^c^ was significantly greater than EV_MSC-T_^c^ (72.7 ± 3.2% vs. 62.4 ± 4.8%, Figure 2C). We next analyzed the expression of cell surface receptors on Mφ1 differentiated with or without EVs and stimulated with lipopolysaccharide (LPS). Results showed that EV treatment reduced the expression of the co-stimulatory molecules CD80 and CD86, and the expression of HLA-DR, in LPS-stimulated Mφ1 to levels resembling those observed for Mφ2 (Figure 2D and Appendix A). When we performed the opposite experiment and added EVs to monocytes differentiating to Mφ2, we failed to detect changes in the pattern of Mφ1 or Mφ2 markers (Figure 2C,D), indicating that EVs appear to only affect Mφ1→Mφ2 differentiation. Analysis of two common pro- and anti-inflammatory cytokines secreted by Mφ during their differentiation to Mφ1 showed that the addition of EVs significantly decreased the levels of the pro-inflammatory cytokine TNF-α in the culture medium after LPS activation, and at the same significantly increased the levels of the anti-inflammatory cytokine IL-10 (Figure 2E). The level of IL-10 secreted by differentiated Mφ1 was significantly higher in the presence of EV_MSC-T-HIF_^C^ than in the presence of EV_MSC-T_^C^. Accordingly, the TNF-α/IL-10 ratio was significantly lower in Mφ1 cultures differentiated in the presence of EVs. Overall, these data demonstrate that EVs can repolarize Mφ1 to functional Mφ2-like. Analysis of Mφ viability revealed no changes in response to EV_MSC-T-HIF_^C^ or EV_MSC-T_^C^ (Appendix A).

### 2.4. EV_MSC-T-HIF_^C^ Enhance the Efferocytic and Immunosuppressive Capacity of Mφ1

Our results show that MSC-EVs can redirect the phenotype of Mφ1 to Mφ2-like. We sought to confirm these results by analyzing the interaction of EV-treated Mφ with other immune cell populations using two in vitro functional tests: neutrophil efferocytosis and peripheral blood mononuclear cell (PBMC) immunosuppression (Figure 3A). Resolution of inflammation is an active process that not only halts pro-inflammatory signaling, but also clears apoptotic polymorphonuclear neutrophils at the injury site [23], a process termed efferocytosis. It has been previously shown that Mφ2 have a greater efferocytic capacity than Mφ1 [24]. We tested the effect of MSC-EVs on efferocytosis (see scheme in Figure 3A, top), finding that the ability of Mφ1 to clear apoptotic neutrophils was significantly greater when they were first differentiated in the presence of EVs, reaching the phagocytic levels of differentiated Mφ2 (Figure 3B). In addition, it is known that when Mφ1 activation diminishes and Mφ show a regulatory/suppressive phenotype, tissue homeostasis is restored in part by the suppression of T-cell proliferation [25]. We developed a T-cell proliferation assay with co-cultures of Mφ1 or Mφ2 differentiated as described (Figure 3A). Results showed that the proliferation of T-cells was significantly lower when co-cultured with Mφ2 than with Mφ1 (Figure 3C). Moreover, whereas the immunosuppressive capacity of Mφ1 differentiated in the presence of EV_MSC-T_^C^ was significantly inferior in comparison with Mφ2, monocytes differentiated in the presence of EV_MSC-T-HIF_^C^ showed an immunosuppressive capacity very similar to Mφ2 (Figure 3C).

### 2.5. EV_MSC-T-HIF_^C^ Suppress Inflammation and Inhibit PBMC Adhesion on Activated Endothelium

Endothelial cells are major participants and regulators of inflammatory reactions. The adhesion of circulating leukocytes to vascular endothelium is a pivotal step for their extravasation during inflammation [26]. We next aimed to evaluate how EV_MSC-T-HIF_^C^ affect adhesion and extravasation processes on activated endothelium. Human umbilical vein endothelial cells (HUVECs) were treated with TNF-α and IL-1β to simulate an inflammatory state and were cultured, or not, with EV_MSC-T_^C^ or EV_MSC-T-HIF_^C^ (Figure 4A). After 24 h of stimulation, we analyzed the expression pattern of genes related to endothelial inflammation, including P-selectin (SELP), integrins (VCAM and ICAM) and metalloproteinase 2 (MMP2). Results showed that the expression of the inflammatory-related genes in HUVEC was lower when cells were co-cultured with MSC-EVs (Figure 4B). Moreover, the expression of VCAM and SELP was significantly lower in the presence of EV_MSC-T-HIF_^C^ than in the presence of EV_MSC-T_^C^, and there was also a trend for a decrease in ICAM and MMP2 expression (Figure 4B). Because inflammation is known to disrupt the cohesion and organization of intercellular junctions, we examined adherens junctions using immunofluorescence staining against VE-cadherin (VE-cad) and platelet endothelial cell adhesion molecule (PECAM) [27]. In the inflammatory environment, the addition of MSC-EVs to HUVEC monolayers could partially restore VE-cad (Figure 4C) and PECAM (Appendix A) architecture, as revealed by an increase in immunofluorescence intensity, and EV_MSC-T-HIF_^C^ seemed to work better than EV_MSC-T_^C^. Given these findings, we next examined immune cell adhesion on activated endothelium. PBMCs were stained with CFSE and were added to HUVEC monolayers for 2 h (Figure 4D). The percentage adhesion of PBMCs to activated HUVECs in the absence of EVs was considered as 100%. We observed residual binding to non-activated endothelium (51.96 ± 5.8%, Figure 4E). Under inflammatory conditions, the adhesion of PBMCs to activated endothelium was significantly lower in the presence of MSC-EVs (Figure 4E). In line with the results obtained for the expression and integrity analysis, the loss of PBMC adhesion was significantly greater with EV_MSC-T-HIF_^C^ than with EV_MSC-T_^C^ (84.8 ± 2.3% vs. 94.1 ± 3.7%) (Figure 4E).

### 2.6. EV_MSC-T-HIF_^C^ Ameliorate Fibrosis Induced by TFG-β Treatment

Fibrosis typically accompanies chronic inflammatory processes induced by a variety of stimuli, including autoimmune reactions, and is attributed to excess deposition of ECM components, including collagen [28]. To test whether MSC-EVs impact fibrosis in an inflammatory context, we activated dermal fibroblast cultures with TGF-β and treated them concomitantly with EV_MSC-T_^C^ or EV_MSC-T-HIF_^C^ for 24 h. Western blot analysis showed that TFG-β treatment increased the steady-state levels of alpha-smooth muscle actin (α-SMA) and collagen type I alpha (COL-1α) (proteins related to fibrosis), and this induction was inhibited by the addition of MSC-EVs (Figure 5A), with EV_MSC-T-HIF_^C^ showing a more pronounced effect. We confirmed these results by immunofluorescence analysis of α-SMA (Figure 5B).

### 2.7. EV_MSC-T-HIF_^C^ Attenuate TNBS-Induced Colitis in Mice

The evident beneficial effects of EV_MSC-T-HIF_^C^ in vitro motivated us to test their therapeutic potential in a model of inflammatory disease. Specifically, we utilized the TNBS-induced mouse colitis model of experimental Crohn’s disease [29,30]. Balb/c mice were previously randomized into three groups: a sham group, a non-treated TNBS group and an EVs-treated TNBS group. Fifty micrograms of EV_MSC-T-HIF_^C^ or PBS were intraperitoneally administered 6 h after acute TNBS-induced colitis and potential anti-inflammatory effects were evaluated 4 days later. Mice in the TNBS group showed reduced colon length compared with sham-treated mice, whereas mice treated with EV_MSC-T-HIF_^C^ after TNBS did not show colon shortening (Figure 6A). Colon histology revealed severe mucosal damage in the TNBS group treated with PBS, characterized by a decrease in the number of intestinal glands, crypt distortion and profuse inflammatory cell infiltration. By contrast, the EV_MSC-T-HIF_^C^-treated TNBS group showed preserved tissue architecture with significantly less histopathological damage (Figure 6B). We also analyzed the expression of inflammatory cytokines and macrophage markers in colon tissue to explore the mechanisms responsible for the alleviation of the symptoms by MSC-EVs. As shown in Figure 6C, the mRNA expression levels of pro-inflammatory cytokines (TNF-α, IL-1β, IL-6) and Mφ1-associated genes (CD86, iNOS and CCR7) were significantly lower in the EV-TNBS-treated group than in the PBS-TNBS group, whereas the opposite was seen for the anti-inflammatory cytokine IL-10. Analysis of Mφ2-associated genes (ArgI and CD206) showed a trend for an increase in the EV-TNBS-treated group. These data indicate that EV_MSC-T-HIF_^C^ can suppress the inflammatory response in TNBS-induced colitis by regulating the expression of cytokines. 

To corroborate these findings, we studied the effect of EV_MSC-T-HIF_^C^ on the phenotype of infiltrated Mφ. Immunofluorescence analysis of colon samples revealed a greater presence of pro-inflammatory Mφ1 (F4/F80^+^-PD-L1^+^) than immunomodulatory Mφ2 (F4/F80^+^-CD206^+^) in the PBS-TNBS group. The opposite result was seen for mice treated with EV_MSC-T-HIF_^C^ (Figure 7A). To further investigate the mechanisms underlying colitis recovery following treatment with EV_MSC-T-HIF_^C^, we used Sirius Red to detect tissue collagen fibers content. Intestinal fibrosis occurs in the context of inflammation, leading to tissue damage and altered tissue reconstruction with excessive ECM deposition [28]. As expected, we found that the percentage of collagen in the colon was significantly higher in the TNBS group than in the sham group (26.8 ± 2.3 vs. 20 ± 0.8), and was significantly lower in the EV-TNBS-treated group (18.5 ± 3.5 vs. 26.8 ± 2.3) (Figure 7B), suggesting that EV_MSC-T-HIF_^C^ treatment alleviated intestinal fibrosis in this model of colitis.

## 3. Discussion

MSCs are excellent candidates for cell therapy owing to their immunomodulatory and regenerative features. It has recently been shown that the EVs released by MSCs can reproduce the beneficial effects of the cells from which they originate [31,32]. EV therapy has advantages over conventional cell therapy, as it can be formulated as an “off-the-shelf” product [33]. Dental pulp is an easily available source of MSCs. In addition to their robust immunoregulatory potential, dental pulp-derived MSCs have a very high proliferative capacity that make them an excellent source of EVs [34].

EVs have key roles in intercellular communication by delivering biological information in the form of chemokines, cytokines, growth factors, lipids and miRNAs, among others [35,36], which can change the fate and properties of recipient cells [37]. In the present study, we show a functional interaction between MSC-derived EVs and immune cells, which accords with other studies using different EV sources [38,39]. Similar to these studies, we observed strong uptake of EVs by monocytes and a noticeable uptake by B-cells. In contrast to the findings of Rutman et al. [38], we also noted a very high uptake by neutrophils, which correlates with the fact that monocytes and neutrophils are the main phagocytic cells [40].

The stimuli that MSCs receive from the environment can modulate their EV cargo and kinetics of release, potentially enhancing their immunoregulatory and regenerative properties [41,42]. Likewise, modifications to the donor cell can alter the functional properties of the secreted EVs [43]. Hypoxia is a physiological characteristic of immunological niches [44], and changes to MSCs driven by HIF-1α can modulate innate and adaptive immunity, controlling the proliferation, differentiation and function of different immune cells [13,45]. Hypoxic preconditioning of MSCs is known to enhance the immunosuppressive properties of secreted EVs [46,47]. We show here that EV_MSC-T-HIF_^C^ exhibits stronger immunoregulatory and anti-inflammatory properties than EV_MSC-T_^C^, which is supported by the following in vitro results: (I) superior differentiation of monocytes to Mφ2-like; (II) weakened adhesion of PBMCs to stimulated endothelium accompanied by partial restoration of the architecture of non-inflamed endothelium and reduced recruitment of immune cells to the tissues; (III) control of excessive fibrosis caused by impaired resolution of inflammation.

Previous studies have shown that EVs derived from preconditioned MSCs have a greater capacity to modulate the M1/M2 phenotype [48,49]. In this regard, we found that the M1/M2 switch can occur during differentiation of monocytes into Mφ. In the aforementioned studies, EVs were added to differentiated THP1 and differentiated Mφ, respectively, which is different to our method where EVs are added during monocyte differentiation and repolarize Mφ1 to phenotypical and functional Mφ2-like. We believe this is relevant as circulating monocytes migrate to injured areas and, as progenitors of Mφ, represent the first link for intervention.

Hypoxia preconditioning can enhance the angiogenic properties of MSC-EVs [14,15,50,51]. However, the effect of hypoxia preconditioning of MSC-EVs on endothelial adhesion and integrity under inflammatory conditions has been less well studied. Merino et al. recently showed that membrane particles derived from adipose tissue MSC have regenerative effects on endothelial cells [52], and Lopatina et al. reported that EVs released by PDGF-stimulated adipose tissue stem cells in vitro enhance PBMC adhesion to endothelium [53]. Here, we found that the expression of integrins and other adhesion molecules on activated endothelial cells was reduced after treatment with both EV_MSC-T-HIF_^C^ and EV_MSC-T_^C^ (but consistently better using the former), resulting in poorer adhesion of PBMCs. In addition, EV_MSC-T-HIF_^C^ downregulated the expression of MMP-2, an important proteolytic for immune cell extravasation [54], and increased the expression of adherens junction proteins on HUVECs, which would reduce immune cell extravasation.

TGF-β signaling drives fibrosis in many immune disorders, including inflammatory bowel disease [55]. A very recent study reported that administration of hypoxic-conditioned EVs could noticeably reduce the expression of collagen-1 and α-SMA, and help preserve airway inflammation and fibrosis in asthmatic mice [56]. Consistent with this, we found that the expression of collagen-1 and α-SMA were more significantly reduced in TGF-β-stimulated fibroblasts cultured with EV_MSC-T-HIF_^C^ than with EV_MSC-T_^C^, pointing to the beneficial effects of HIF-1 α genetic modification for the use of EVs in ameliorating fibrosis.

Crohn’s disease is a relapsing chronic inflammatory bowel disease with two fundamental mechanisms associated with its pathogenesis: mesenchymal–epithelial transition and inflammatory cell infiltration. This results in the release of inflammatory cytokines and remodeling of the ECM [57]. MSC therapy has been investigated for inflammatory diseases such as Crohn’s disease [58]. Indeed, an advanced MSC therapy (Darvadstrocel) is approved in the EU for the treatment of complex perianal fistulas in Crohn’s disease [59], and several clinical trials are investigating the potential of MSC-derived EVs to ameliorate Crohn’s disease pathology [60]. TNBS-induced colitis recapitulates some aspects of Crohn’s disease, including a significant immune response induced by IL-12 released by Th1 cells [61]. Th1 cells also produce IFN-γ, which polarizes recruited monocytes to an M1 phenotype, resulting in an increase in pro-inflammatory cytokines such as TNF-α, IL-1β and IL-6 [62]. Encouragingly, TNBS-treated mice administered with EV_MSC-T-HIF_^C^ showed both a decrease in pro-inflammatory cytokine expression and an increase in the expression of the anti-inflammatory cytokine IL-10. IL-10 regulates excessive pro-inflammatory responses by inhibiting the ability of antigen-presenting cells to present antigens to T-cells and is largely produced by Mφ2 [63]. We suggest that the healing effect of EV_MSC-T-HIF_^C^ on colitis is at least in part due to the skewing of the Mφ1/2 phenotype. 

Intestinal inflammation often leads to fibrosis, which can be observed in the TNBS-induced colitis model [64]. In this regard, we demonstrate that EV_MSC-T-HIF_^C^ treatment reduces the excess fibrosis generated by TNBS.

Overall, our results show that EV_MSC-T-HIF_^C^ can interact with immune cells, endothelium and fibroblasts, and modify the pro-inflammatory environment to an inflammation-resolving and tissue healing–promoting state. Accordingly, EV_MSC-T-HIF_^C^ might have therapeutic utility for autoimmune disorders such as Crohn’s disease.

## 4. Materials and Methods

### 4.1. Ethical Statements

All donors gave their informed consent for inclusion. The study was conducted in accordance with the Declaration of Helsinki, and the protocol was approved by the Ethics Committee of The Hospital La Fe Universitari i Politècnic, Valencia, Spain (Project identification 2019/0101).

Animal procedures were approved by Hospital La Fe Ethics Committee (Protocol N° 2021/VSC/PEA/0060) according to guidelines from Directive 2010/63/EU of the European Parliament on the protection of animals used for scientific purposes.

### 4.2. Human Samples

Human dental pulp samples were obtained from third molars, which were extracted for orthodontic reasons from healthy young people (18–21 years of age).

Buffy coats of healthy donors were obtained from the Blood Bank, (Hospital Universitari i Politècnic La Fe, Valencia, Spain) after informed consent.

### 4.3. Cell Culture

Human dental pulp MSCs were expanded and transduced with the lentiviral vectors pLV-hTERT-IRES-hygro (Addgene, 85140; Watertown, MA, USA) and pWPI-HIF-1α-GFP, as described [65,66]. Transduction efficiency was evaluated by hygromycin selection and flow cytometry and the percentage of infection was typically ~90%. Validation of non-senescence was previously demonstrated [16]. MSCs were cultured in Dulbecco’s modified Eagle’s medium (DMEM)-low glucose (Gibco, Thermo Fisher Scientific, Waltham, MA, USA) supplemented with 10% heat-inactivated fetal bovine serum (FBS, Corning, Glendale, AZ, USA) and 100 U/mL penicillin and 100 μg/mL streptomycin (P/S, Sigma-Aldrich, San Luis, MO, USA). To obtain EVs, MSCs were cultured in extraction medium (EM), which was prepared by supplementing DMEM with 10% EV-depleted FBS and antibiotics. EVs-depleted FBS was generated by ultracentrifugation of regular FBS and DMEM mixed 1:1 at 100,000× *g* for 16 h. MSCs were cultured in conditioning medium by adding IFN-γ (50 ng/mL, R&D, Minneapolis, MN, USA), TNF-α (10 ng/mL, R&D) and IL-1β (10 ng/mL, Peprotech, London, UK) to complete EM. Primary cultures of HUVEC were obtained from Lonza and were grown in Endothelial Cell Growth Medium-2 (EGM-2) BulletKit™ (Lonza, Basel, Switzerland). HUVEC were stimulated with TNF-α (10 ng/mL, R&D) and IL-1β (10 ng/mL, R&D) during 24 h in the presence, or not, of 15 µg/mL MSC-EVs. Fibroblasts were isolated in the laboratory from skin biopsies [67] and were cultured in DMEM/F12 (Gibco, Thermo Fisher Scientific) supplemented with 10% FBS and 1% P/S. One day before stimulation, fibroblasts were seeded in DMEM/F12 serum-free medium supplemented with 1% P/S. Fibroblasts were stimulated with TGF-β (10 ng/mL, R&D) during 24 h in the presence, or not, of 15 µg/mL of MSC-EVs in DMEM/F12 serum-free medium supplemented with 1% P/S.

PBMCs and neutrophils were isolated by density gradient centrifugation with Histopaque (Sigma-Aldrich, Darmstadt, Germany) and Lympholyte-M (Tebu-bio, Yvelines, France), respectively. PBMCs were cultured in Rosewell Park Memorial Institute (RPMI) medium (Gibco, Thermo Fisher Scientific) supplemented with 10% FBS (Corning), 1 mM pyruvate, 2 mM glutamine and 1% P/S (all from Sigma-Aldrich). Monocytes were isolated by positive magnetic separation (Miltenyi Biotech, Bergisch Gladbach, Germany). CD14^+^ cells (1 × 10^6^) were cultured in 24-well flat-bottom culture plates for 6 days in complete RPMI (Gibco, Thermo Fisher Scientific) supplemented with 10% FBS, 1 mM pyruvate, 2 mM glutamine and 1% P/S (all from Sigma-Aldrich). To generate monocyte-derived Mφ1 or Mφ2, 5 ng/mL recombinant human granulocyte macrophage-colony stimulating factor (rhGM-CSF, Peprotech) or 20 ng/mL recombinant human macrophage-colony stimulating factor (rhM-CSF, Peprotech) were added to complete medium, respectively. Cytokine stimulation was repeated on days 2 and 4. During the last 16 h of culture 10 ng/mL of lipopolysaccharide (LPS, Invitrogen, Waltham, MA, USA) was added as needed. During Mφ1 or Mφ2 differentiation, 15 µg/mL of EVs were added on day 0 of culture.

### 4.4. EVs Isolation and Characterization

We used ultracentrifugation without a sucrose-gradient centrifugation step to isolate EVs from MSCs. EVs were isolated from approximately 100 mL of EM after 48 h of incubation of ~2 × 10^7^ MSCs using several ultracentrifugation steps, as described [16]. EV protein concentration was determined with the Pierce BCA Protein Assay Kit (Thermo Fisher Scientific) to ensure equal amounts of protein samples for experiments. EVs were suspended in RIPA buffer (1% NP40, 0.5% deoxycholate, 0.1% sodium dodecyl sulphate in Tris-buffered saline (TBS), Sigma-Aldrich) for Western blotting or in PBS for characterization and functional analysis. NTA and electron microscopy were performed as described [68]. Pre-embedding immunogold staining was performed by incubating sections in the primary antibody anti-CD63 (Abcam, TS63, Cambridge, UK) and then in colloidal gold-conjugated secondary antibody (anti-IgG mouse, Abcam).

### 4.5. Uptake of Labeled EVs

To measure EV uptake by different immune cell population, first EVs were labeled with CFSE (Thermo Fisher Scientific) as previously described [69]. CFSE stock was made following the manufacturer’s instructions by adding 18 µL of dimethyl sulfoxide (DMSO, Sigma-Aldrich) to the CFSE dye resulting in a 5 mM stock. In order to stain EVs with CFSE, 1 µL of CFSE stock was added to 30 µg of EVs diluted in 1mL PBS and incubated for 15 min at 37 °C in darkness in a 1 mL tube. After this time, the EVs were transferred to an ultracentrifuge tube with 20 mL of filtered PBS. Unincorporated dye was removed by an ultracentrifugation step. EVs were resuspended in 30 µL of filtered PBS and added to 100,000 PBMCs and neutrophils seeded in a p24 well. CFSE mixed with PBS was used as a negative control to normalize the amount of unincorporated dye. After 3 h of incubation, CFSE-positive cells were detected by flow cytometry. EV uptake in different immune cells was compared and analyzed using cell-specific surface markers.

### 4.6. Flow Cytometry

Cells were first incubated with a blocking solution (PBS containing 1% of normal mouse serum) for 10 min and then incubated with saturating amounts of fluorochrome-conjugated antibodies for 1 h at 4 °C. Human antibodies used were: anti-CD14 (RPE, Dako, TUK4, Santa Clara, CA, USA), anti-CD3 (PerCP-Cy, BD Biosciences, SK7), anti-CD4 (BV510, BD Biosciences, L200), anti-CD8 (PE-Cy7, BD Biosciences, RPA-T8), anti-CD19 (APC-H7, BD Biosciences, SJ25C1), anti-CD56 (Alexa Fluor 647, BD Biosciences, B159), anti-CD15 (FITC, Biolegend, HI98,), anti-CD163 (PerCP-Cy, BD Biosciences, GHI/61), anti-CD80 (APC, BD Biosciences, FUN-1), anti-CD86 (V450, BD Biosciences, L307.4) and anti-HLA-DR (FITC, Miltenyi Biotec, AC122) at concentrations recommended by the manufacturers. Results were analyzed using a BD FACSCANTO II flow cytometer equipped with FlowJo® software (FlowJo LLC, BD, Franklin Lakes, NJ, USA).

### 4.7. Cytokine Measurements

Levels of TNF-α and IL-10 in culture supernatants were measured by enzyme-linked immunosorbent assays (BioLegend, San Diego, CA, USA).

### 4.8. Cell Viability

Differentiated Mφ1 or Mφ2 treated or not treated with EVs were detached from the plastic with trypsin/EDTA (Gibco), and were washed and resuspended in binding buffer with annexin-V-FITC and propidium iodide (R&D) for 15 min. The percentage of viable, apoptotic and dead cells was then measured by flow cytometry.

### 4.9. Efferocytosis 

After isolation, neutrophils were cultured for 24 h to induce apoptosis. Neutrophils were stained with Texas Red (Thermo Fisher Scientific) and co-cultured with LPS-stimulated Mφ (1:1 ratio) for 2 h. Monocytes differentiated to Mφ1 were treated or not treated with EVs on day 0 of differentiation. Cells were stained with anti-human CD14 (RPE, Dako, TUK4) and analyzed by flow cytometry. Efferocytosis was scored as the percentage of CD14^+^ cells (Mφ) that were also APC^+^, having phagocytosed stained apoptotic neutrophils.

### 4.10. T-cell Proliferation Assay

T-cell proliferation was examined as described [70]. PBMCs were labeled with 5 μM (CFSE) and activated with Dynabeads™ Human T-Activator CD3/CD28 (Thermo Fisher Scientific). LPS-stimulated Mφ were co-cultured with 1 × 10^5^ CFSE-labeled and activated PBMCs to evaluate their immunosuppressive potential. Monocytes differentiating to Mφ were treated or not treated with EVs. After 5 days of culture, PBMCs were analyzed by flow cytometry. First, we gated on CD3^+^ population and then proliferation of T-cells was determined using the CFSE dilution method [71]. At day zero the CFSE staining is high but as cells activate and expand the CFSE intensity is diluted. If T-cell activation is inhibited, the CFSE intensity does not dilute as much and remains at a higher intensity. The expansion index (EI) was estimated using FlowJo software. Percentage of immunosuppression was calculated by normalizing data to a 0–100% scale by establishing 0% immunosuppression for the EI of Mφ1 and 100% immunosuppression for the EI of non-activated PBMC using the following formula:$$Immunosuppression\;\left(\%\right)=\frac{EI^{M\Phi 1}-EI^{sample}}{EI^{M\Phi 1}-EI^{non-act}}\times 100$$

### 4.11. PBMC Adhesion 

HUVECs were cultured at high confluence and stimulated with TNF-α (10 ng/mL, R&D) and IL-1β (10 ng/mL, Peprotech) for 24 h treated or not treated with 15 µg/mL EVs at the same time as the stimulation. In total, 1 × 10^5^ CFSE-labeled PBMCs were placed on HUVEC monolayers. After 2 h, non-adherent cells were removed and monolayers were washed twice with PBS. The number of attached PBMCs was measured by the fluorescence intensity value at 492^EX^/517^EM^ nm of each well using a fluorescence plate reader. Images were acquired with 10× objective on a Paula Cell Imager (Leica Microsystems, Wetzlar, Germany) combining phase contrast and green fluorescence.

### 4.12. Western Blotting

EVs or cells were lysed in RIPA buffer containing protease (Complete, Sigma-Aldrich) and phosphatase (PhosSTOP, Sigma-Aldrich) inhibitors. Equal amounts of protein samples were mixed with non-reducing Laemmli sample buffer (BioRad, Hercules, CA, USA) and denatured at 96 °C for 5 min. Proteins were separated on 10% SDS-polyacrylamide gels and transferred to polyvinylidene difluoride membranes (Immobilon-P, Millipore, Burlington, MA, USA). Membranes were blocked with TBS containing 5% (*w*/*v*) non-fat dry milk powder. Human primary antibodies used for Western blotting were: anti-HSP70 (dilution 1/500, Cell Signaling Technology, D69, Danvers MA, USA), anti-TSG101 (dilution 1/200, Santa Cruz, C-2, Dallas, TX, USA), anti-CD9 (dilution 1/500, Santa Cruz, C-4), anti-calnexin (dilution 1/1.000, Santa Cruz, H-70), anti-α-SMA (dilution 1/200, Dako, clone 1A4), anti-COL-1α (dilution 1/500, Cell Signaling, E8I9Z) and anti-tubulin (dilution 1/1.000, Sigma-Aldrich, T5168). Secondary antibodies used were anti-IgG rabbit (dilution 1/4.000, Dako, P0448) and anti-IgG mouse (dilution 1/10.000, Sigma-Aldrich, A9044). Detection was carried out using peroxidase-conjugated secondary antibodies and the ECL Plus Reagent (GE Healthcare, Chicago, IL, USA) or SuperSignal^TM^ West Femto (Thermo Fisher Scientific). Reactions were visualized using an Amershan Imager 600 (GE Healthcare) and quantified with ImageJ software (NIH, Bethesda, MD, USA).

### 4.13. Real Time Quantitative PCR

RNA was extracted using RLT buffer (Qiagen, Dusseldorf, Germany) and purified with the RNeasy Plus Mini Kit (Qiagen). RNA was quantified spectrophotometrically using a NanoDrop ND-1000 (NanoDrop Technologies, Wilmington, DE, USA). cDNA was obtained by reverse transcription using the PrimeScript RT Reagent Kit (Takara, Kusatsu, Japan). RT-qPCR was performed with the respective human-specific sense and antisense primers and RT-SYBR™ Green PCR Master Mix (Applied Biosystems). Multiwell plates of 384 wells were run on a Viia 7 PCR System (Applied Biosystems). The designed primers used were: 

hGAPDH CCCCTCTGCTGATGCCCCA (F) and TGACCTTGGCCAGGGGTGCT (R);

hVCAM AGGTGACGAATGAGGGGACCACA (F) and CCAGCCTCCAGAGGGCCACT (R);

hICAM CTGGTCCTGCTCGGGGCTCT (F) and GGGCTGGTCACAGGAGGTGC (R);

hSELP TTGTTCCTGCCAGCAGCTGCC (F) and AGGGCCAGAGACCCGAGGAG (R);

mActin GCCAACCGTGAAAAGATGACC (F) and GAGGCATACAGGGACAGCAC (R)

mTNF CCCTCACACTCAGATCATCTTCT (F) and GCTACGACGTGGGCTACAG (R);

mIL-1β GAAATGCCACCTTTTGACAGTG (F) and CTGGATGCTCTCATCAGGACA (R);

mIL-6 GAGTCCTTCAGAGAGATACAGAAAC (F) and TGGTCTTGGTCCTTAGCCAC (R);

mCD86 GCACGGACTTGAACAACCAG (F) and CCTTTGTAAATGGGCACGGC (R);

miNOS CGCTTGGGTCTTGTTCACTC (F) and GGTCATCTTGTATTGTTGGGCTG (R);

mCCR7 CTCTCCACCGCCTTTCCTG (F) and ACCTTTCCCCTACCTTTTTATTCCC (R);

mIL-10 GGACAACATACTGCTAACCGAC (F) and CCTGGGGCATCACTTCTACC (R);

mArgI GTGGGGAAAGCCAATGAAGAG (F) and TCAGGAGAAAGGACACAGGTTG (R);

mCD206 TGTGGAGCAGATGGAAGGTC (F) and TGTCGTAGTCAGTGGTGGTTC (R).

### 4.14. Immunofluorescence of Human Cell Cultures

Cells were fixed and permeabilized with cold 70% ethanol solution and human antibodies used for staining were the following: mouse anti-human VE-Cad (Santa Cruz, sc-9989), mouse anti-human PECAM-1 (Santa Cruz, sc-71872) and mouse anti-human α-SMA-Cy3 (Sigma, C6198). We used goat anti-mouse IgG (1:1000, Alexa Fluor^®^ 488, Abcam) as a secondary antibody. Nuclei were stained with DAPI. Quantification of MFI was performed using ImageJ. Each point shown in the graph corresponds to the mean of 50 cells.

### 4.15. Mice

Adult male Balb/c mice (6 weeks old, 22−26 g) were purchased from Charles River Laboratories Inc., (Wilmington, MA, USA), and maintained under standard laboratory conditions. All animal procedures were approved by institutional ethical and animal care committees.

### 4.16. TNBS-induced Colitis

TNBS colitis was induced by an intrarectal administration of 100 µL of TNBS (3.5 mg per 20 g mice, Sigma-Aldrich) dissolved in 40% ethanol, as previously described [30]. Vehicle-treated mice received 100 µL of 0.9% NaCl dissolved in 40% ethanol. Some mice were treated 6 h after TNBS induction intraperitoneally with 50 µg of EVs in 100 µL of PBS or only PBS. Mice were killed by cervical dislocation on day 4 after TNBS administration. Colon length was measured and colon tissue was frozen in liquid nitrogen for RNA extraction and fixed in 4% paraformaldehyde acid and embedded in paraffin for immunohistochemistry.

### 4.17. Mouse Histology and Immunofluorescence

Paraffin-embedded colon samples were cut into 5 µm-thick sections and stained with hematoxylin-eosin (Sigma-Aldrich) to evaluate inflammatory infiltrates, the presence of ulceration and the lesion of crypts. A Picro-Sirius Red stain (Direct Red 80 and Picric Acid, Sigma-Aldrich) was used to visualize fibrosis tissue. Slides were visualized on a Leica DMD108 Digital Microscope (Leica Microsystems). For immunofluorescence, slides were blocked with 5% normal goat serum and 0.1% Triton X-100 in PBS for 1 h. Slides were incubated with rat anti-F4/F80 (dilution 1/200, Abcam, ab6640), rabbit anti-CD206 (dilution 1/200; Abcam, ab64693) or rabbit anti-CD274 (dilution 1/200, AB Clonal A11273) overnight in a humidified chamber at 4 °C and then slides were washed three times for 5 min each in PBS. Slides were incubated with anti-rat IgG Alexa 555 or anti-rabbit IgG Alexa 488 secondary antibodies for 1 h and then washed three times for 5 min in PBS. Cell nuclei were stained with DAPI and slides were mounted using FluorSave™ Reagent (Merck Millipore). The sections were observed and visualized under fluorescent microscope Leica DM2500 (Leica Microsystems). Final image processing and quantification were performed using ImageJ software by counting green and red spots in the fixed area.

### 4.18. Statistical Analysis

Data are expressed as mean ± SD or SEM, as specified. Student’s *t* test was used for unpaired samples in between-group comparisons. One-way analysis of variance (ANOVA) was used to compare means of more than 2 groups. Two-way ANOVA was used to simultaneously evaluate the effect of two factors on a response variable. Analyses were conducted with GraphPad Prism 8 software (San Diego, CA, USA). Differences were considered statistically significant at *p* < 0.05 with a 95% confidence interval.

## 5. Conclusions

In conclusion, EVs isolated from MSCs cultured under conditions resembling an inflammatory state can create a local microenvironment favorable for tissue healing processes. We demonstrate that EV_MSC-T-HIF_^C^ can interact with major immune cell players of resolution of inflammation such as Mφ1, and both endothelial cells and fibroblasts, providing a potential “cell-free” therapy for the treatment of immunological and inflammatory diseases such as Crohn’s disease.

## Figures and Tables

**Figure 1 ijms-22-11269-f001:**
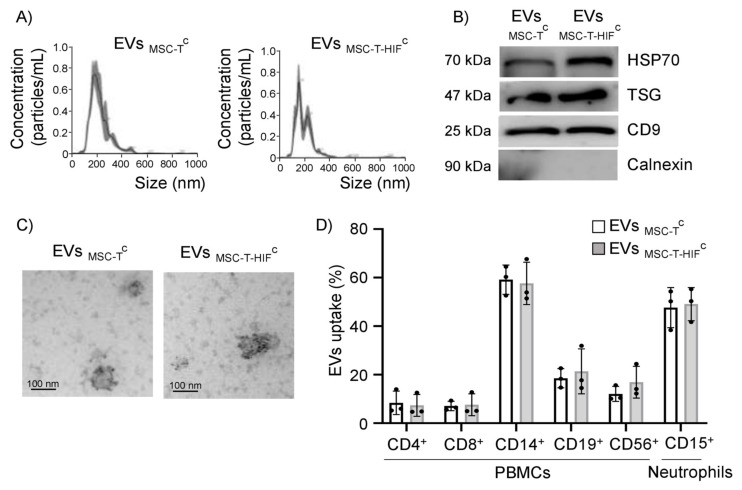
Characterization of MSC-EVs and their uptake by different leukocyte subpopulations. (**A**) Representative images of EVs assessed by nanoparticle tracking analysis; (**B**) representative Western blots of Hsp70, TSG101 and CD9 proteins in 30 µg of loaded of EVs; absence of calnexin signifies a pure EV preparation; (**C**) representative transmission electron microscopy images of isolated EVs. Scale bar: 100 nm. EVs showed positive immunogold-labeling of the transmembrane extracellular protein CD63, a canonical EV marker. (**D**) Peripheral blood mononuclear cells and neutrophils were incubated with carboxyfluorescein succinimidyl ester (CFSE)-labeled EVs for 3 h at 37 °C and EV internalization was assessed by flow cytometry. As a negative control, PBS was mixed with CFSE and added to immune cells in parallel. Cell types were analyzed by specific surface markers for T-cells (CD4^+^ and CD8^+^), B-cells (CD19^+^), monocytes (CD14^+^), natural killer cells (CD56^+^) and neutrophils (CD15^+^). EV internalization was measured by fluorescence intensity and is represented as the percentage of EVs internalized by each cell population. Graphs represent mean ± SD of three independent experiments.

**Figure 2 ijms-22-11269-f002:**
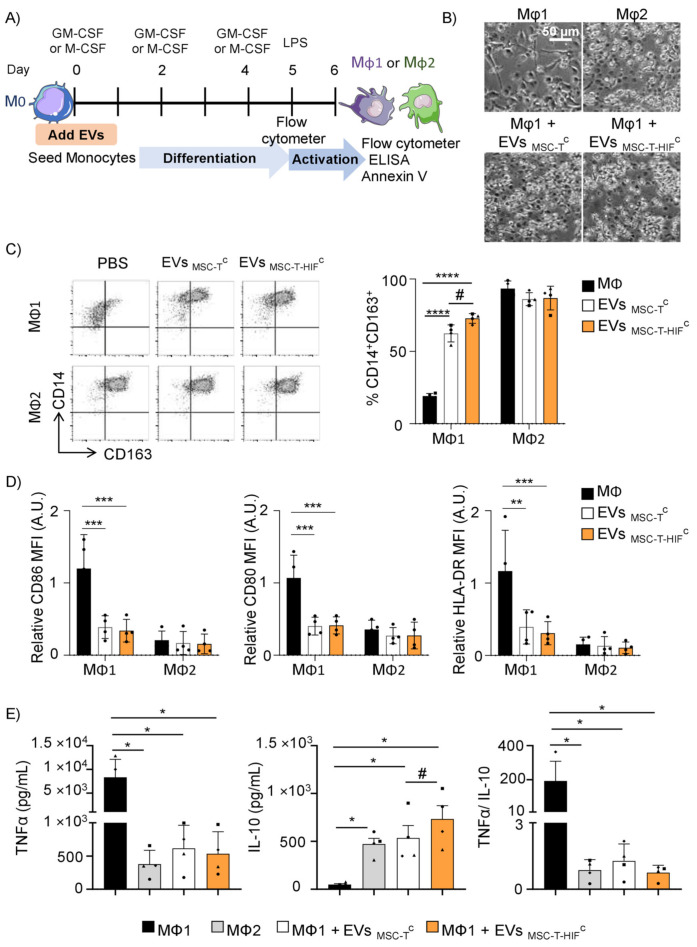
EV_MSC-T-HIF_^C^ are superior to EV_MSC-T_^C^ in repolarizing Mφ1 to an Mφ2-like phenotype. (**A**) Scheme of the in vitro experimental design; (**B**) brightfield microscopy images of Mφ after five days of differentiation. Scale bar: 50 µm. (**C**) After 5 days of differentiation, the percentage of CD14^+^ and CD163^+^ cells were assessed by flow cytometry. Representative dot plots are shown. Black bars represent Mφ1 or Mφ2 without EVs, respectively. Graph represents mean ± SD of four independent experiments. Two-way ANOVA was used for statistical analysis. (**D**) Sixteen hours after lipopolysaccharide (LPS) activation, CD86, CD80 and HLA-DR expression was assessed by flow cytometry. Relative mean fluorescence intensity (MFI) was calculated by dividing all individual data by the mean expression in Mφ1. Black bars represent Mφ1 or Mφ2 without EVs, respectively. Graphs represent mean ± SD of four independent experiments. Two-way ANOVA was used for statistical analysis. (**E**) TNF-α and interleukin IL-10 production by Mφ was determined by ELISA 16 h after LPS stimulation. Black bars represent Mφ1 and gray bars Mφ2 without EVs, respectively. Graphs represent mean ± SD of four independent experiments. One-way ANOVA with Geisser–Greenhouse correction was used for statistical analysis. Each symbol (●, ■, ▲, ♦) represents a different paired experiment. * ^or #^ *p* < 0.05, ** *p* < 0.01, *** *p* < 0.001, **** *p* < 0.0001.

**Figure 3 ijms-22-11269-f003:**
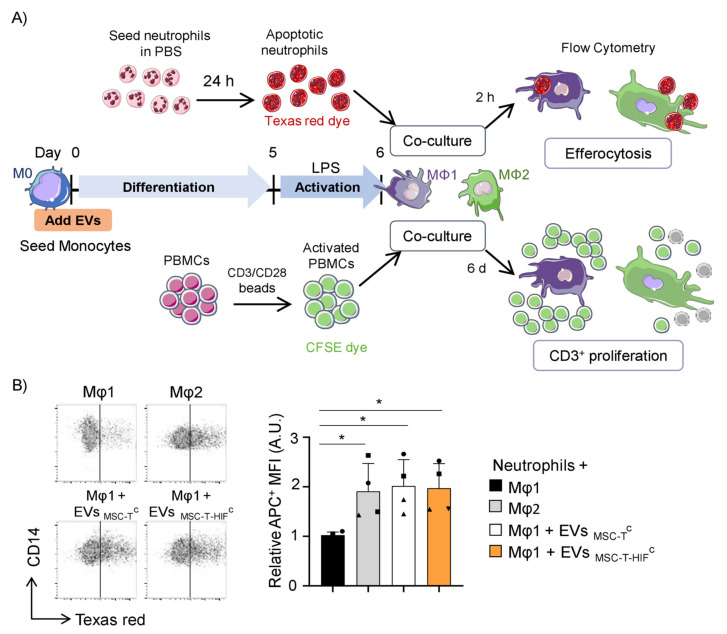
Efferocytic and immunosuppressive capacity of Mφ treated with EV_MSC-T-HIF_^C^. (**A**) Scheme of the in vitro experimental design; (**B**) apoptotic neutrophil internalization was measured by increase in the APC channel and represented as mean fluorescence intensity (MFI) in each condition. Relative MFI was calculated by dividing all individual data by the MFI of Mφ1. Graph represents mean ± SD of four independent experiments. One-way ANOVA with Geisser–Greenhouse correction was used for statistical analysis. Representative dot plots are shown. (**C**) Proliferation of T-cells. Peripheral blood mononuclear cells were stained with carboxyfluorescein succinimidyl ester (CFSE) and stimulated with anti-CD3 and anti-CD28 beads before co-culture with differentiated Mφ1 treated or not treated with EVs, or with differentiated Mφ2. After 5 days, cells were stained with anti-CD3 antibodies and proliferation of T-cells was determined by flow cytometry measurement of CFSE dilution. Suppression (percentage) was calculated based on the expansion index. Graph represents mean ± SD of four independent experiments. One-way ANOVA with Geisser–Greenhouse correction was used for statistical analysis. Each symbol shape (●, ■, ▲, ♦) represents a different paired experiment. * *p* < 0.05.

**Figure 4 ijms-22-11269-f004:**
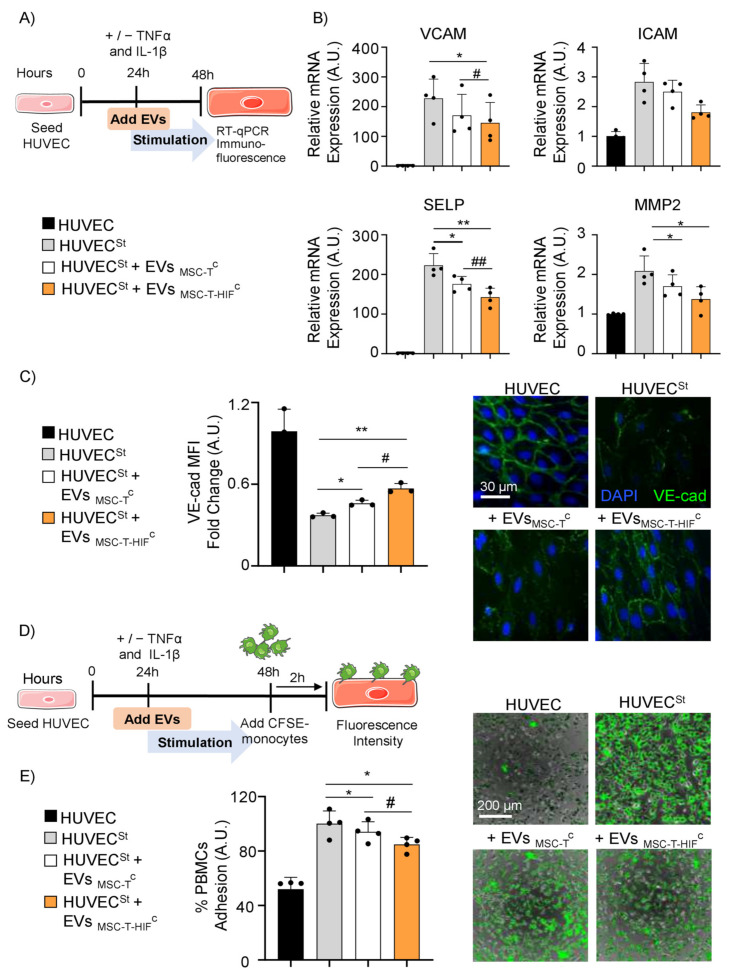
EV_MSC-T-HIF_^C^ reduce inflammatory responses and peripheral blood mononuclear cell (PBMC) adhesion to activated endothelium. (**A**) Scheme of the in vitro experimental design; (**B**) VCAM, ICAM, SELP and MMP2 expression levels quantified by RT-qPCR in TNF-α- and IL-1β-stimulated HUVEC (HUVEC^St^). HUVEC^St^ were treated with EV_MSC-T_^C^ and EV_MSC-T-HIF_^C^. Unstimulated HUVECs were used as a control. Expression level of the target gene in each sample was normalized to GAPDH expression. Graphs represent mean ± SD of fold change of four independent experiments. One-way ANOVA with Geisser–Greenhouse correction was used for statistical analysis. (**C**) Immunofluorescence of VE-cadherin (VE-cad, green) and nuclei staining (blue) to show the distribution of VE-cad in the cell membrane. Scale bar: 30µm. Bar graph shows quantification of mean fluorescence intensity (MFI). Relative MFI was calculated by dividing all individual data by the MFI in unstimulated HUVEC. Graph represents mean ± SD of fold change of three independent experiments. One-way ANOVA with Geisser–Greenhouse correction was used for statistical analysis. (**D**) Scheme of the in vitro experimental design. (E) Images of PBMCs (green) adhering to unstimulated HUVEC and HUVEC^St^ during 2 h, treated or not treated with EV_MSC-T_^C^ or EV_MSC-T-HIF_^C^. Scale bar: 200 µm. Percentage of PBMC adhesion was calculated considering the stimulated condition as 100% adhesion. Graph represents mean ± SD of fold change of four independent experiments. One-way ANOVA with Geisser–Greenhouse correction was used for statistical analysis. * ^or #^ *p* < 0.05, ** ^or ##^ *p* < 0.01.

**Figure 5 ijms-22-11269-f005:**
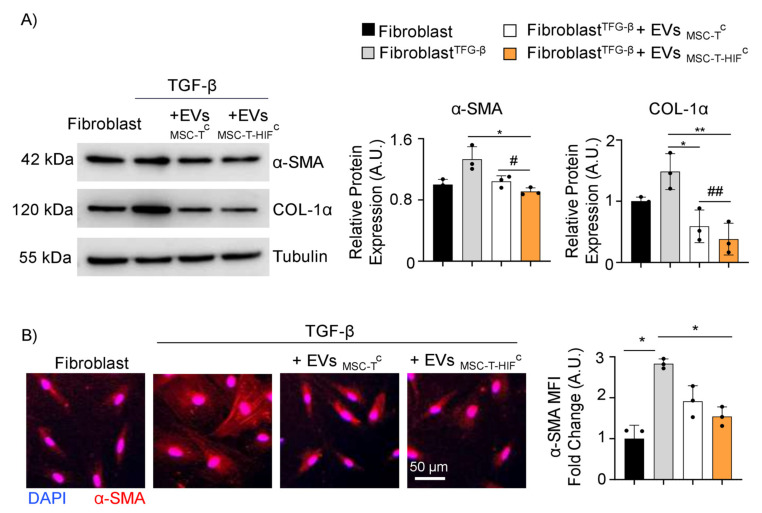
EV_MSC-T-HIF_^C^ ameliorate fibrosis induced by TGF-β. (**A**) Representative Western blots of α-SMA and COL-1α in unstimulated and TFG-β-stimulated dermal fibroblasts treated with EV_MSC-T_^C^ or EV_MSC-T-HIF_^C^. Expression levels were quantified by densitometry relative to the levels in unstimulated fibroblasts. Tubulin was used as a loading control. Graphs represent the mean ± SD of three independent experiments. One-way ANOVA with Geisser–Greenhouse correction was used for statistical analysis. (**B**) Immunofluorescence of α-SMA (red) and nuclei staining (blue). Scale bar: 50 µm. Bar graph shows quantification of red mean fluorescence intensity (MFI). Relative MFI was calculated by dividing all individual data by the MFI in unstimulated fibroblasts. Graph represents the mean ± SD of three independent experiments. One-way ANOVA with Geisser–Greenhouse correction was used for statistical analysis. * ^or #^ *p* < 0.05, ** ^or ##^ *p* < 0.01.

**Figure 6 ijms-22-11269-f006:**
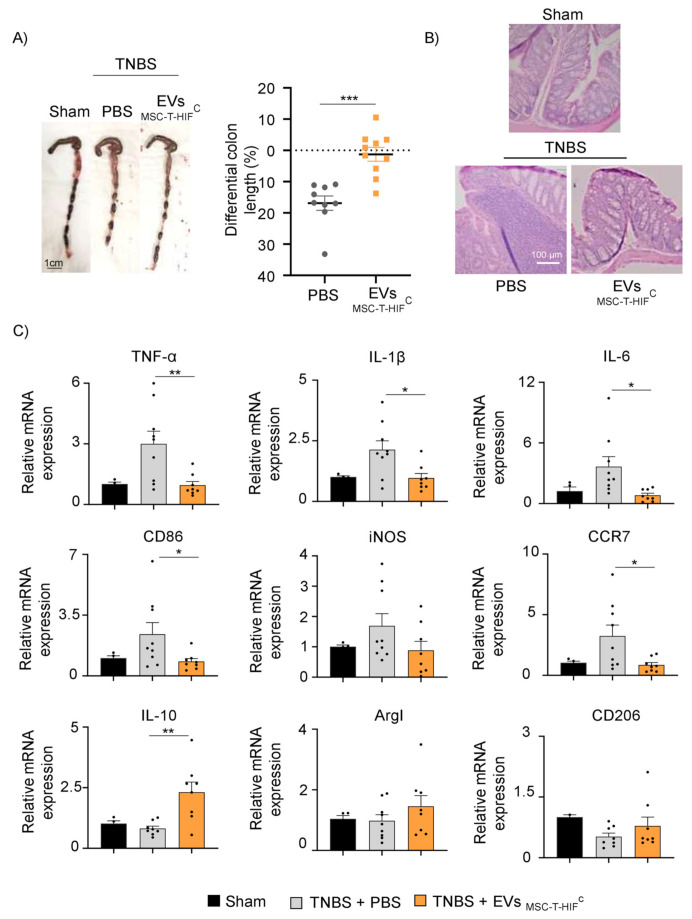
EV_MSC-T-HIF_^C^ attenuate disease in mice with TNBS-induced colitis. (**A**) Macroscopic images of colonic tissues at day 4 after TNBS administration. Scale bar: 1 cm. Percentage of differential colon length compared with the sham group (horizontal dot line). Data are presented as the mean ± SEM of 9 mice in each group. Unpaired *t*-test was used for statistical analysis. (**B**) Hematoxylin and eosin staining of representative histological sections of mouse colon in the sham group and in the PBS and EV groups after TNBS administration. Scale bar, 100 μm. (**C**) TNFα, IL1-β, IL-6, CD86, iNOS, CCR7, IL-10, CD206 and ArgI mRNA expression levels quantified by RT-qPCR in mouse colon in the sham group and in the PBS and EV groups after TNBS administration. Sham group was used as control. Expression level of the target gene in each sample was normalized to β-actin expression. Graphs represent mean ± SEM of fold change of nine independent experiments. Unpaired *t*-test was used for statistical analysis. * *p* < 0.05, ** *p* < 0.01, *** *p* < 0.001.

**Figure 7 ijms-22-11269-f007:**
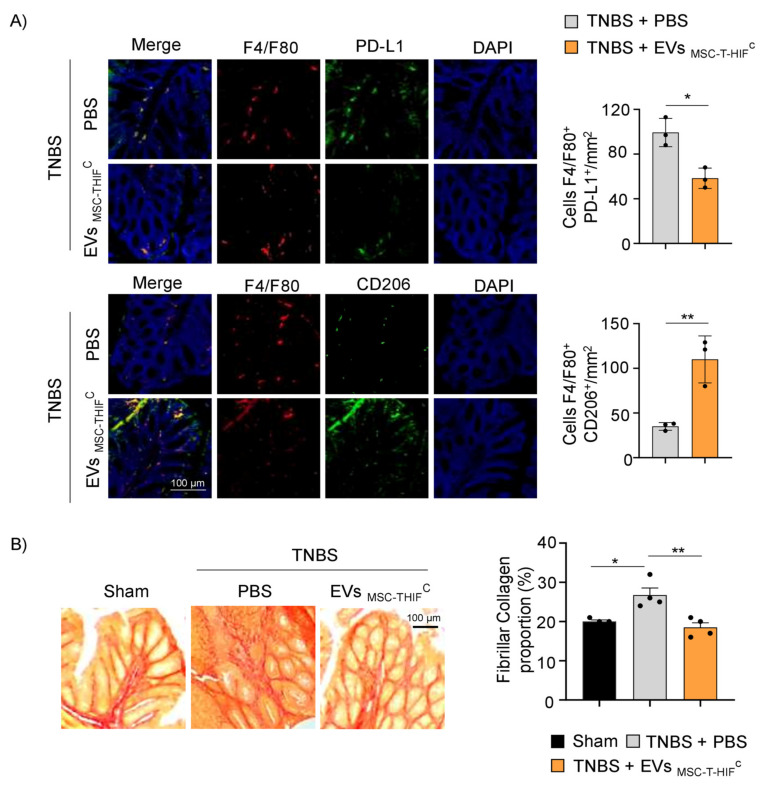
EV_MSC-T-HIF_^C^ change the ratio of infiltrating Mφ1/2 and decrease fibrillar collagen content. (**A**) Immunodetection of F4/F80 (pan-macrophage marker, red) and PD-L1 (pro-inflammatory, Mφ1, green) or CD206 (Mφ2, green) in colon samples 4 days after TNBS-induced colitis. Scale bar: 100 μm. Quantification of double-positive cells per mm^2^. Ten sections of 0.14 mm^2^ per mouse were analyzed. Graphs represent the mean ± SEM of three mice. Unpaired *t*-test was used for statistical analysis. (**B**) Sirius Red staining was used to detect collagen fibers. Fibrillar collagen content (%) was calculated by dividing red stained area by total tissue area. Scale bar: 100 μm. Graph represents mean ± SEM of four mice. Unpaired *t*-test was used for statistical analysis. * *p* < 0.05, ** *p* < 0.01.

## Data Availability

All data generated and/or analyzed during this study are included in this published article and its additional files. All the data can be shared upon request by email.

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
