# Peer review of "HIF-Overexpression and Pro-Inflammatory Priming in Human Mesenchymal Stromal Cells Improves the Healing Properties of Extracellular Vesicles in Experimental Crohn’s Disease"

_ijms, 2021, doi:10.3390/ijms222011269_

Round 1

Reviewer 1 Report

In the article entitled “HIF-overexpression and pro-inflammatory priming in human mesenchymal stromal cells improves the healing properties of extracellular vesicles in experimental Crohn’s disease” by Gómez-Ferrer and colleagues, the authors show that extracellular vesicles derived from MSCs have therapeutic potential for the treatment of Crohn’s disease. The authors have carried out many different experiments including in vitro and in vivo models. The design of the experiments is well thought out. The text is nice to read, the figures are clear. Statistical methods are adequate. The work makes a good impression.

Minor questions and comments.

  1. Scale bar is absence on some representative microscopy images including Figures 2B, 4C, 4E, 5B, 6A and Supplementary Figure 3.
  2. Figure 5B quality is not so good.
  3. Why do authors use SEM on the Figures 6 and 7? SD is showed on the other figures.

Author Response

Minor questions and comments.

1. Scale bar is absence on some representative microscopy images including Figures 2B, 4C, 4E, 5B, 6A and Supplementary Figure 3.

You are right, thank you very much for your appreciation. All the scale bars have been incorporated both in the microscopy images and in the captions.

2. Figure 5B quality is not so good.

Thanks for the comment, often the quality of the images decreases a lot when converting the ppt files to Tiff extension. However, we have improved the resolution and included new images.

3. Why do authors use SEM on the Figures 6 and 7? SD is showed on the other figures.

When using animals, variability is often high within the experimental group. Thus, in animal experiments, the standard error is usually the most useful of the statistics that measure variability to reduce uncertainty due to noise. Some guidelines for experimental design and statistical analyses in animal studies recommends the presentation of the pooled SEM because the objective in comparisons of treatments in animals is usually to provide inferences about the population. In these cases, the dispersion of the sample mean should be well considered, because the mean value is the key characteristic that differs between study groups. The size of the sample is then directly related to the soundness of the scientific inference (doi: 10.1201/9781315155807, doi:10.5713/ajas.18.0468 and doi: 10.1097/JP9.0000000000000024).

But if you are not satisfied with this statistical analysis we can change the error bars of figures 6,7 to standard deviation.

Reviewer 2 Report

This paper describes the effects of extracellular vesicles derived from MSCs engineered to overexpress hypoxia-inducible factor 1-alpha and telomerase (MSC-T-HIF) and conditioned with pro-inflammatory stimuli

The experimental procedures are well controlled, and the described results are valid. The data are mostly presented appropriately. The methods are only in part well described

In particular, the following points should be addressed.

Major comments:

  1. The schemes in the figures 2, 3,4 are clear and well describe the experimental design but the imagine resolution should be improved
  2. T cell proliferation assay is not convincing or not well described. Were T cells analysed by cytofluorimetric analysis using an anti-CD3 antibody?
  3. A representative cytofluorimetric imagine of CFSE method should be added in all figures describing this method
  4. The method of uptake of labelled EVs should be better described (which concentration of CFSE? Which well,flask,tube do you use for the labeling? Is this method validated? The authors should give evidence of the controls used and also here,they should show a representative analysis in the figures
  5. Why the authors use one way ANOVA with Geisser-Greenhouse correction for statistical analysis?

Minor comments:

In the figure 2 on the top Gomez-Ferrer, et al, Figure 2 is to delete. Moreover, are the authors sure that the magnification in the panel B is 100X?

Author Response

Major comments:

1. The schemes in the figures 2, 3,4 are clear and well describe the experimental design but the imagine resolution should be improved

Thank you for your appreciation. We have improved resolution of all figures.

2. T cell proliferation assay is not convincing or not well described. Were T cells analysed by cytofluorimetric analysis using an anti-CD3 antibody?

Proliferation of T-cells was determined by flow cytometry using the CFSE dilution method. This versatile dye is membrane permeant, so it enters into the cytoplasm where its acetate groups are removed by cellular esterases. Then, CFSE stably binds to the abundant amine groups present in cytoplasmic molecules, conferring a stable fluorescence intensity to cells which is equally divided between daughter cells after each division. With each division the CFSE labelling is diluted 1/2. (doi: 10.1016/bs.mie.2019.05.020 and doi: 10.1046/j.1440-1711. 1999.00877.x.)

We used PBMCs to develop the assay. Within the lymphocyte population, CD3+ T cells contribute to the most significant portion of PBMCs (45-70%). Moreover, we used Dynabeads® human T-lymphocyte activator CD3/CD28 for T-lymphocyte activation and expansion. The beads are covalently bind with anti-CD3 and anti-CD28 antibodies. These two antibodies provide primary and co-stimulatory signals optimized for efficient T-cell activation and expansion. Thus, after 5 days of activation the majority of the PBMC culture is enriched with T lymphocytes (CD3+, CD28+). However, to analyze the CFSE dilution we first gated on the CD3 population by staining the PBMCs with the anti-CD3 antibody and then the Incidence of Expansion is measured on that population.

We have improved material and methods (paragraph “T cell proliferation assay”) to clarify this experiment.

3. A representative cytofluorimetric imagine of CFSE method should be added in all figures describing this method

Thank you for your suggestion. We have included a representative cytofluorimetric imagine helps to understand the outcome of the experiment in Figure 3C.

In figure 4 the CFSE was measured by plate reader so we cannot show cytofluorimetric imagine of CFSE this result.

4. The method of uptake of labelled EVs should be better described (which concentration of CFSE? Which well,flask,tube do you use for the labeling? Is this method validated? The authors should give evidence of the controls used and also here, they should show a representative analysis in the figures

Staining EVs to be used in uptake experiment is a well-established technique in the field of EVs (doi: 10.1111/sji.12371, 10.1016/bs.mie.2020.09.002, 10.1038/s41598-017-01731-2 and 10.3892/ijmm.2014.1663). We have completed materials and methods explaining the technique further.

Supplementary figure 1 show representative analysis. In this figure the first plots are the control of the experiment. PBS stained with 5 µM of CFSE were washed with ultracentrifuge and then added to the immune cells. The bars in figure 1D show the capture percentages by subtracting the percentage obtained with PBS.

5. Why the authors use one way ANOVA with Geisser-Greenhouse correction for statistical analysis?

We have used Geisser-Greenhouse correction because it is what literature and GraphPad handbook recommends (https://www.graphpad.com/guides/prism/latest/statistics/stat_sphericity_and_compound_symmet.htm): If your experiment design is repeated measures (multiple measurements over time), they don´t recommend to assume sphericity. When you can’t assume sphericity, you have to correct and the recommended method is Greenhouse-Geisser correction when epsilon is less than 0.75, which is our case. (Greenhouse, S. W.; Geisser, S. (1959). "On methods in the analysis of profile data". Psychometrika24: 95–112.)

Minor comments:

In the figure 2 on the top Gomez-Ferrer, et al, Figure 2 is to delete. Moreover, are the authors sure that the magnification in the panel B is 100X?

Thank you for noticing the top part of the image, it has already been deleted.

The magnification is 100x by the addition of the magnification of the objective lens (10x) and the eyepiece lens (10x).

Round 2

Reviewer 2 Report

The manuscript has been improved as suggested by the reviewers.